# Physical Characterization of Bismuth Oxide Nanoparticle Based Ceramic Composite for Future Biomedical Application

**DOI:** 10.3390/ma14071626

**Published:** 2021-03-26

**Authors:** Pravin Jagdale, Gianpaolo Serino, Goldie Oza, Alberto Luigi Audenino, Cristina Bignardi, Alberto Tagliaferro, Carlos Alvarez-Gayosso

**Affiliations:** 1Center for Sustainable Future Technologies—IIT@PoliTO Istituto Italiano di Tecnologia (IIT), Via Livorno 60, 10144 Turin, Italy; pravin.jagdale@iit.it; 2Department of Applied Science and Technology (DISAT), Politecnico di Torino, Corso Duca degli Abruzzi 24, 10129 Turin, Italy; alberto.tagliaferro@polito.it; 3Department of Mechanical and Aerospace Engineering (DIMEAS), Politecnico di Torino, 10129 Turin, Italy; alberto.audenino@polito.it (A.L.A.); cristina.bignardi@polito.it (C.B.); 4Polito^BIO^Med Lab, Politecnico di Torino, 10129 Turin, Italy; 5National Laboratory for Microfluidics and Nanofluidics (LABMyN), Centro de Investigación y Desarrollo Tecnológico en Electroquímica (CIDETEQ), 76703 Queretaro, Mexico; goza@cideteq.mx; 6Laboratorio de Investigacion de Materiles Dentales, Division de Estudios de Posgrado e Investigacion, Facultad de Odontologia, UNAM, Circuito de la Investigacion Cientifica, Alcaldia de Coyoacan, Ciudad Universitaria, 04510 Mexico, Mexico; calvarezgayosso@comunidad.unam.mx

**Keywords:** epoxy, bismuth oxide, nanoparticles, composite, biocompatible, mechanical properties, nanoindentation, macro–micro mechanical characterization

## Abstract

Employment and the effect of eco-friendly bismuth oxide nanoparticles (BiONPs) in bio-cement were studied. The standard method was adopted to prepare BiONPs-composite. Water was adopted for dispersing BiONPs in the composite. A representative batch (2 wt. % of BiONPs) was prepared without water to study the impact of water on composite properties. For each batch, 10 samples were prepared and tested. TGA (thermogravimetric**** analysis) performed on composite showed 0.8 wt. % losses in samples prepared without water whereas, maximum 2 wt. % weight losses observed in the water-based composite. Presence of BiONPs resulted in a decrease in depth of curing. Three-point bending flexural strength decreased for increasing BiONPs content. Comparative study between 2 wt. % samples with and without water showed 10.40 (±0.91) MPa and 28.45 (±2.50) MPa flexural strength values, respectively, indicating a significant (*p* < 0.05) increase of the mechanical properties at the macroscale. Nanoindentation revealed that 2 wt. % without water composites showed significant (*p* < 0.05) highest nanoindentation modulus 26.4 (±1.28) GPa and hardness 0.46 (±0.013) GPa. Usage of water as dispersion media was found to be deleterious for the overall characteristics of the composite but, at the same time, the BiONPs acted as a very promising filler that can be used in this class of composites.

## 1. Introduction

In the emerging field of biotechnology, there has been a constantly increasing development of smarter biocompatible materials. Development of these materials aimed at improving their bioactivity and interactions with cells [1,2]. Materials like metal, polymer, ceramic, glass, etc., are largely used in repairing and restoration of teeth [3] and bones [4]. Polymers and cement are well appreciated in in-vivo and in-vitro biomedical applications due to their aesthetic, injecting, and molding abilities [5,6,7]. During the past fifty years, polymer-cement composites have been studied intensively [8,9,10]. Bio-cements were introduced in 1960 [11,12] for biomedical applications specially in reconstruction of bony defects and to stabilize implanted devices [13,14].

Factors that influence the properties and biocompatibility of composite are onsite curing duration and rate [15], heat exchange [16], depth of polymerization [17], and fillers [18,19]. These properties were continuously explored to enhance thermo-chemical properties [20,21] and biocompatibility of the composite [22]. The number of monomeric chains in the matrix is mainly responsible for the polymeric properties [23]. Introduction of reduced size (nano/micro) bioactive species able to move through the polymeric lattice enhances the beneficial effect of cements used in the oral cavity or for bone reconstruction. These kinds of biocomposites must bear the mechanical stress and abrasion to avoid the leaching of toxic species, from the polymeric lattice, during their service life [24]. It is therefore mandatory to investigate the mechanical response of composites on which depends the success or the failure of implants and reconstructions.

In recent years, high performance nano- and micro-sized [22] organic and inorganic materials were developed for biomedical applications [25,26]. Zinc, magnesium, calcium, aluminum, and silicate tune the strength, bioactivity, photopolymerization, settling time, and temperature with other crucial aspects [27]. Calcium and phosphate enhance the biocompatibility property by improving the restoration and the stability of implants over time [13,27]. Bismuth based materials also grabbed attention due to their: cost effective, eco-friendly properties (bismuth element and its compounds have very low toxicity) [28], biocompatibility, and easy production [29]. These properties of bismuth based material raised a lot of interest in biomedical applications [30]. Furthermore, bismuth oxide (Bi_2_O_3_) is already studied as a disinfectant, bacteriostatic agent, astringent [31], and as a radiopaque element in cement used for endodontic application [32]. Furthermore, it is also explored in the various applications like functional ceramics [33], catalysts [34], pharmaceuticals [35], sensor [36], etc.

In literature, bismuth oxide nanoparticles (BiONPs) with various morphologies (particles, belt, rods, tubes, fibers, dots, etc.) were reported [37]. Numerous techniques such as precipitation, microwave, electrodeposition, metal organic chemical vapor deposition, laser pulsed deposition, cathodic pulverization, hydrothermal, and thermochemical [29,37] have been used to produce BiONPs.

In the reported work, thermo-chemical technique was adopted in order to have a low cost, easy and quick synthesis of the desired shape and size of BiONPs [29]. Furthermore, a multiscale approach to define the mechanical properties of the composites was adopted. As reported in other studies the nanoindentation technique in recent years was widely applied to characterize different kinds of materials, from biomaterials to synthetic ones [38,39], also in conjunction with tests performed at higher scale level [38,40].

Composite properties are mainly influenced by the size, concentration, dispersion, and mutual interaction of nano/micro fillers in the matrix [30], the impact of BiONPs on the shrinkage of the composite was studied in detail [41] but, to the author’s knowledge, no studies were performed to understand how the BiONPs impact the structural properties of cement composite at different scale levels. Standard dentistry restoration cement was used as a blank sample which was useful to set the benchmark for comparison of composite properties. The null hypothesis here tested was that the structural properties of the composite cement were not affected by the filler and water.

This study demonstrated that the use of BiONPs as a filler in composite cements complies with the required depth of curing threshold. Adding BiONPs to the cement enhances the nanoindentation elastic modulus and hardness. Better results were achieved by 2 wt. % of BiONPs than 1 wt. % and 4 wt. % of BiONPs, showing that the optimized amount is close to that value. Hence, a reasonable concentration of BiONPs can enhance the structural properties of bio cement. Previous works suggested that the addition of water can promote the sealing ability of the cement [42]. However, in our case, the addition of water decreased the mechanical properties of the BiONPs based composite. For this reason, we prepared an additional batch of composite with optimized concentration (2 wt. %) of BiONPs without water [43]. This allowed us to highlight the detrimental influence of water/humidity on composite curing and structural properties [37]. Anyway, further investigations, testing other filler concentrations, are needed to deeply understand the impact of the BiONPs on the structural properties of the restoration cements.

## 2. Materials and Methods

### 2.1. BiONPs Preparation

Bismuth-nitrate penta-hydrate salt (99%), produced in our laboratory, was used as a precursor for the synthesis of BiONPs. In-house developed solid state thermoxidation reaction (SSR) [29] was employed to synthesize BiONPs. The oxidation of precursor was performed under oxygen (50 mL/min) atmosphere at 500 °C for 30 min in a quartz reactor. The reactor cooled down to atmospheric temperature. The obtained agglomerated BiONPs were finally crushed in powdered form in mortar and pestle. This step was followed to ensure well separation and free flowing BiONP powder.

### 2.2. Composite Preparation

The nanoparticles, in the form of plates, are highly agglomerated. Hence, we adopted a mortar–pestle procedure to break the BiONPs agglomeration into a fine powder followed by sonication in water to keep the BiONPs suspended without agglomeration before adding into the composite preparation process. The composites were prepared by dispersing BiONPs in quantities of 1, 2, and 4 wt. %. Deionized water 10, 20, and 40 wt. % (respectively for the following percentage quantities of BiONPs: 1, 2, and 4 wt. %) was used to achieve better dispersion of BiONPs in the composite. Water was used as a solvent due to its biocompatibility. The quantity of BiONPs was calculated with respect to the weight of silica dioxide. Water quantity was considered with respect to the quantity of bisphenol-A glycerolate dimethacrylate (Bis-GMA from Sigma-Aldrich, St. Louis, MO, USA). Initially, BiONPs–water mixture was stirred at 1200 rpm for 5 min. Bisphenol-A glycerol dimethacrylate (10.50 g) was added and stirred at 1200 rpm for 10 min. Next, 4.50 g of tri(ethylene glycol) di-methacrylate (Sigma-Aldrich, St. Louis, MO, USA), 0.05 g of 1, 2—camphor quinone (Sigma-Aldrich, St. Louis, MO, USA), and 0.20 g of N, N-dimethyl-p-toluidine (Sigma-Aldrich, St. Louis, MO, USA) were mixed and stirred at 800 rpm for 15 min. The process followed stirring at each stage to maintain the suspension and uniform size distribution of heavy BiONPs in the mixture. Silica dioxide (Sigma-Aldrich, St. Louis, MO, USA) in a quantity of 35.58 g was added finally with vigorous stirring. The composite mixture was molded in required dimensions and cured under irradiance between ultraviolet (UVA) and blue visible light (385 and 515 nm) with power density of 1000 mW/cm^2^ for 40 s through the curing unit Bluephase C8 (Ivoclar Vivadent, Liechtenstein). The cured samples were further kept in atmospheric conditions for 24 h to complete the polymerization. To study the impact of water on composite properties, another batch of 2 wt. % of BiONPs composite was prepared without using water. BiONPs (0.7 g) were added in Bis-GMA and stirred at 1200 rpm for 10 min. The other steps for sample preparation remained the same. For each concentration, 10 samples were prepared (Table 1) using the appropriate mold.

Table 2 summarizes the chemical elements used to prepare the composite.

### 2.3. Depth of Cure

Depth of polymerization or curing is a preliminary requirement for any biopolymer or composite designed for in-vivo applications [44]. In order to evaluate the curing efficiency of the composite, i.e., the depth of curing, the specimens were prepared according to ISO 4049 standards [45]. Measurements of depth of curing were performed with a specific experimental setup shown in Figure 1. A re-usable stainless-steel mold (6 mm high and 4 mm diameter) was positioned on a glass plate, already covered with a transparent mylar film. The mold was filled with the prepared composite resin and then covered with another mylar film and glass plate. The flatness of the upper and lower surfaces was obtained by gently pressing the upper glass plate. Glass slide from the top was removed and the composite in the mold was irradiated using a visible blue light (1000 mW/cm^2^) for 40 s. The light intensity was adjusted and monitored with a curing radiometer (Demetron Research Corp, Danbury, CT, USA). The cured composite specimen was taken out from the mold. The absolute length of the cured cylindrical specimens was measured with a vernier caliper. Ten samples for each concentration of BiONPs were prepared. Means and standard deviations were calculated.

### 2.4. Three-Point Bending Test

A custom steel mold (25 × 2 × 2 mm^3^) was used to prepare the samples to be tested through the three-point bending procedure. After dripping the composite in the mold, the curing procedure was performed on both sides under visible blue light at 1000 mW/cm^2^ (Figure 2) with three overlapping light irradiations for 40 s each. After curing, the samples were exposed in a humid chamber for 15 min. Finally, all the samples were kept in water for 24 h. The measurements were performed according to the ISO 4049 standards [45]. The 3-point bending tests were performed by using the universal mechanical test machine (Instron 5567, Norwood, MA, USA). The distance between the support pins was fixed at 20 mm. The crosshead speed was set to 1 mm/min with a pre-load of 0.5 N.

The stress and strain were calculated according to the following equations:σ = (3FL)/(2bh^2),(1)
ε = (6bδ)/(L^2),(2)
where F is the force, L is the distance between the supports (20 mm), b is the specimen width, h is the height of the rectangular bar, and δ is the deflection.

The failure point is described by the flexural strength (σ_fs_) that was calculated through Equation (1) when the force reached the maximum value.

The flexural modulus was calculated from the linear elastic region of the stress–deflection curve using:E = (F′L^3)/(4bh^3δ′),(3)
where δ′ is deflection corresponding to the maximum load (F′) in the linear portion of the force-deflection curve. Mean value and standard deviation were calculated on the data obtained from 10 samples of each filler concentration.

### 2.5. Nanoindentation Test

Microstructural elastic modulus and hardness were analyzed through the Nanoindenter XP (MTS, Eden Prairie, MN, USA) equipped with a Berkovich diamond tip and characterized by a theoretical force resolution of 50 nN and a theoretical displacement resolution less than 0.01 nm. To determine the area function of the indenter cross section a fine calibration on a standard silica sample was performed [46]. Microscope to indenter calibration was also performed to properly set the offset distance between the light microscope of the indenter and the indentation site. Here displacement-controlled tests were performed.

Typical nanoindentation curves are characterized by an initial loading phase where the tip of the indenter penetrates the surface, a holding phase where the tip is held in order to stabilize the viscous phenomena, and by an unloading phase where the tip is retracted from the sample.

The loading and unloading phases for these experiments were characterized by a constant strain rate set to 0.1 s^−1^. During the holding phase i.e., when the maximum indentation depth of 2000 nm was reached, the corresponding load was held constant for 30 s.

The load-displacement curves were analyzed by using the Oliver–Pharr method [47]. The nanoindentation modulus (Equation (4)), and the nanoindentation hardness (Equation (5)) values were calculated using the following equations:1/E* = (1 − ν_i_)/E_i_ + (1 − ν)/E,(4)
H = P_max_/A_c_,(5)
where E_i_ (1000 GPa) and ν_i_ (0.07) are, respectively, the Young’s modulus and the Poisson’s ratio of diamond, whereas E and ν are respectively the Young’s modulus and the Poisson’s ratio of the samples. The Poisson’s ratio of the samples was set to 0.3 in accordance with the values defined for these kinds of materials [48]. P_max_ is the maximum indentation load and A_c_ is the contact area. For each concentration of nanofiller, nanoindentation was performed on 40 different locations on the sample’s surface (two samples were sorted for every batch).

### 2.6. Structural Analysis of the Composite

Structural analysis and nanoparticle dispersion in the composite were studied by using field emission scanning electron microscopy (FESEM) with the AG-SUPRA 40 (Carl Zeiss, Jena, Germany). The non-conductive surface of composites was coated with chromium thin layer (5 nm) to obtain charging effect and noise-free images in FESEM.

A transmission electron microscope (TEM) JEM-ARM200F (Jeol, Peabody, MA, USA) was used to examine the physical parameters of the nanoparticles as well as to obtain inter-planar distance in the lattice structure. BiONPs were dispersed in ethanol and, after sonication, collected on a carbon coated copper grid for TEM analysis.

Thermogravimetric analysis (TGA) was performed on all cured samples to study the weight losses. The heating ramp rate of TGA was maintained at 10 °C/min from 25 °C to 120 °C under air atmosphere at a rate of 50 mL/min.

### 2.7. Statistical Analysis

To individuate and discard the outliers, the modified Thompson’s Tau method was applied. To study the influences of filler on composite’s mechanical properties, the one-way analysis of variance (ANOVA) was applied. Two-way analysis of variance was used to understand the simultaneous effect of the dispersion medium and the filler on the mechanical properties of the composite. All the analyses were performed at a 95% confidence level.

## 3. Results and Discussion

### 3.1. BiONPs and Their Interaction in the Composite

BiONPs and clusters were in the sub micrometric size (Figure 3a) with diameters greater than 20 nm. TEM image (Figure 3b) confirmed the well-arranged crystalline nature of the BiONPs. The particles were nanosized in all three dimensions as denoted by the translucency of the plates.

Dispersing BiONPs in cement composites was the major challenge. The secondary electron image of FESEM analysis highlights the BiONPs in the composite (Figure 4). Formation of clusters was observed when 1 wt. % and 4 wt. % of BiONPs are added using water. Instead, using 2 wt. % of filler permitted to obtain better results in terms of dispersion i.e., no clusters were observed (Figure 4c,e).

Indeed, considering that EDS analysis is sensible to the scanned area (of 4 µm^2^), the counterintuitive trend reported in Figure 5a corroborated the hypothesis that the reduction of Bi element (Figure 5a,b) can be addressed to the presence of clusters.

### 3.2. Depth of Curing

Preliminary test for qualification of composite for in-vivo application is measuring the depth of curing. For these types of applications, minimum 2 mm depth of curing is required [49]. The results show that all the prepared samples comply with the required condition by achieving a higher value than 2 mm (Figure 6).

Composite with 1 wt. % BiONPs with water showed similar depth of curing compared with the blank, instead composite with 2 and 4 wt. % BiONPs showed gradual decrease, which can be addressed by the water quantity (10, 20 and 40 wt. %) used for BiONPs dispersion. It evidently confirmed that increasing water concentration after certain limits (1 wt. %) affects the depth of curing.

To confirm it, Figure 6b shows that 2 wt. % composite without water is characterized by the highest depth of curing over all the other samples including the blank one. It indicates that composite without water molecules achieved better polymerization.

### 3.3. Thermal Stability by Thermogravimetric Analysis (TGA)

TGA was performed to study the humidity contained in the composite material (Figure 7). In TGA analysis, the weight losses until 125 °C are due to low volatile species and humidity present in the composite. In Figure 7a, the blank sample showed gradual but slight weight losses (0.8 wt. %). The composites prepared with water, showed the higher weight losses (1.2, 1.4, and 2 wt. %) as compared to the blank sample. The weight losses in the composites showed direct proportionality with quantity of water used for their preparation, as a matter of fact the samples prepared dispersing 4 wt. % of BiONPs showed the highest weight losses (2 wt. %). The composite prepared with 2 wt. % of BiONPs dispersed in water showed a weight loss at 120 °C of 1.3 %. The composites prepared using water, as dispersion media, retain humidity after polymerization.

To understand the impact of water on the thermostability properties, composite samples (2 wt. %) prepared with and without water were compared with the blank ones (Figure 7b). The samples prepared without water showed negligible water losses (0.8 wt. %) as compared to samples prepared with water (1.4 wt. %). The analysis verifies the presence of moisture due to water used in composite preparation, indicating that water molecules entrapped in the cement lattice and voids [50] can influence the physical properties of the composites [51].

### 3.4. Macro and Micro Mechacnial Properties

#### 3.4.1. Three-Point Bending Tests

Representative curves of the three-point bending tests were reported in Figure 8. All the curves are characterized by a very small elastic region, indicating a plastic behavior of the composites. Using BiONPs in quantity of 2 wt. % dispersed in water (Figure 8a) induced a dramatic decrease of the stresses. Increasing the quantity of filler up to 4 wt. %, on the other hand, induced a strengthening of the composites. As a matter of fact, the maximum stress value before the failure increases if compared with that obtained for the cement at 2 wt. % (Figure 8a). Using 1 wt. % of filler produced a little increase of the deformability of the composites. Furthermore, referring to Figure 8b, the absence of water indicates that using 2 wt. % of filler slightly reduced the value of the maximum stress before failure.

Curves reported in Figure 8b indicate that using water, on one hand, increased the deformability of the composite but, on the other hand, dramatically weakened the composite.

The maximum values of stress that the prepared samples can bear before their failure were reported in the histograms of Figure 9. A value of flexural strength of 38.59 (±4.16) MPa was obtained for the blank sample, and values of 33.04 (±7.13) MPa, 10.40 (±0.91) MPa, and 22.02 (±0.94) MPa were measured respectively for samples obtained with 1 wt. %, 2 wt. % and 4 wt. % of BiONPs. Specimens obtained with 2 wt. % of BiONPs without using water resulted in a value of flexural strength of 28.45 (±2.50) MPa.

Resistance of the composites to bending is measured through the flexural modulus reported in the histograms of Figure 10, which results to be 5.05 (±1.19) GPa for the blank specimens and 3.54 (±0.670) GPa, 0.72 (±0.06) GPa, and 1.65 (±0.94) GPa for the watered composite with 1 wt. %, 2 wt. % and 4 wt. % of BiONPs respectively. Flexural modulus for specimens fabricated using 2 wt. % of BiONPs without using water resulted in a mean value of 3.89 (±0.47) GPa.

Generally, the mechanical performance of a composite is critically affected by the bonding between the nanoparticle (or nanoparticles clusters) and the matrix composite [52]. Indeed, the variation of nanoparticle concentration modulates the mean values of flexural strength and flexural modulus: 1 wt. % BiONPs composite slightly weakens the structure. Increasing the BiONPs concentration up to 2 wt. % further reduces the flexural mechanical properties. When the BiONP concentration reaches a value of 4 wt. % the decreasing trend of the registered mechanical properties starts to increase, whereas the value of the flexural strength and flexural modulus still be lower than that obtained for the blank specimens.

As shown elsewhere the coexistence of large nanospheres, in lower percentage than smaller nanofiller, may improve the flexural strength [53]. As a matter of fact, increasing the filler concentration clearly increases the probability of incorporate smaller nanoparticles and strength the cement.

Two phenomena can explain the singular trend obtained for the measured quantity of flexural strength and flexural modulus: (i) packing density of nanospheres and (ii) the absence of crack bridging and crack deflection. Introducing nanoparticles above a certain value (1 wt. %) has not reported any significant influence on the mechanical properties. When the filler reaches a value of 4 wt. % the packing density allows the creation of an adequate number of contact points between the nanoparticles and so the formation of clusters or agglomeration. Adding BiONPs over 2 wt. % increases the probability of crack deflection and crack bridging as demonstrated by the inversion of the decreasing trend of the flexural strength and flexural modulus (Figure 9 and Figure 10). Roughly speaking loading the composite with 4 wt. % of BiONPs increases the probability of occurrence of the aforementioned toughening mechanisms leading to an increase of the flexural properties of the composites [52,54].

The composite prepared without water using 2 wt. % of BiONPs shows an increase in structural mechanical properties as compared to the composite with the same BiONPs concentration but prepared using water (Figure 9b and Figure 10b). It can be speculated that the absence of water molecules permits a better distribution of BiONPs increasing, thus, the likelihood of occurrence of phenomena such as crack deflection and crack bridging that enhances fracture resistance and flexural modulus of the composite at a lower value of BiONPs concentration.

#### 3.4.2. Nanoindentation Analysis

Representative nanoindentation curves obtained for all the specimens were reported in Figure 11. The indentation-load curves indicate that the specimens are characterized by viscous-plastic characteristics. The creep phenomena registered during the holding phase can be addressed to the viscous properties of the samples on the other side the partial recovery of the indentation depth indicates that the composites behave in the same manner as a plastic material. Furthermore, the absence of the nose (i.e., the decrease of the indentation depth at the onset of the unloading phase), permitted to extract the correct mechanical properties [55].

The samples obtained using 2 wt. % of filler were able to withstand to higher load when compared with the other samples (Figure 11a). Furthermore, the presence of water, as already noticed at the macroscale, weakens the composites reducing the load needed to indent the sample (Figure 11b).

The bar chart in Figure 12a shows the mean values of the nanoindentation modulus. The highest value of the nanoindentation modulus for the composites obtained using water and 2 wt % of BiONPs was 12.48 (±1.02) GPa. Respectively for the blank samples and samples obtained dispersing 1 wt. % of BiONPs in water, the values of the mean nanoindentation modulus are 9.50 (±0.61) GPa and 10.40 (±0.60) GPa. A mean nanoindentation modulus value of 11.12 (±1.40) GPa resulted from adding 4 wt. % of BiONPs using water. The statistical analysis indicates that the BiONPs slightly modulate the nanomechanical properties of the composite. Adding 1 wt. % of BiONPs does not modify the nanoindentation modulus properties of the composite (*p* > 0.05).

Composite samples obtained without using water and loaded with 2 wt. % of BiONPs showed a tremendous increase of the nanoindentation modulus reaching a value of 26.82(±1.28) GPa (Figure 12b).

A mild effect of nanoparticle concentration (dispersed in the composites through water) is observed on the nanoindentation hardness (Figure 13a). The obtained values are 0.36 (±0.020) GPa, 0.40 (±0.032) GPa, 0.42 (±0.050) GPa, and 0.42 (±0.12) GPa respectively for the blank samples and for samples obtained loading 1 wt. %, 2 wt. %, and 4 wt. % of BiONPs. Significant differences of nanoindentation hardness values were highlighted by the statistical analysis for samples obtained adding 1 wt. % and 2 wt. % of BiONPs (Figure 13a). The nanoindentation hardness of the composite prepared without water and using 2 wt. % of BiONPs resulted to be 0.46 (±0.013) GPa (Figure 13b).

Obtained values of the nanomechanical properties are coherent with similar dental composites analyzed in other studies [3,54,56,57]. The nanoindentation modulus increases with the filler concentration. An inversion of the trend is observed for the higher value of the filler concentration. A slight reduction of the mechanical characteristics when the filler loading exceeds a certain threshold (4 wt. %) was already observed in literature [57]. Increasing the quantity of nanoparticles rises the probability of clusters formation i.e., at the characteristic scale length of a nanoindentation test the probability to indent in an area of the matrix characterized by a poor concentration of filler increases, explaining the counterintuitive trend in Figure 12a and Figure 13a.

The mechanical characterization performed through nanoindentation technique is very sensitive to the packing density of the filler [58]. For the lowest values of filler concentration, the packing density decreases as well as the contacts between the filler particles and the cement matrix [59] leading to a weakening of the cement. Increasing the filler concentration results in an enhancement of the mechanical properties. The composite can bear higher forces for equal values of deformation, as indicated by the increasing values of the nanoindentation hardness (Figure 13a). Increasing the mechanical properties is strictly correlated to a more homogeneous dispersion of the filler [54,60], so dispersing the BiONPs without water was more effective than dispersing the nanoparticles in water. The lattice voids in this case were filled more efficiently by the nanoparticles.

## 4. Conclusions

Electron microscopy analysis confirmed that the solid-state thermo-chemical method is a useful technique to obtain a homogeneous structure of BiONPs. Due to the low toxicity and eco-friendly properties, BiONPs can be considered as a potential material for bio-applications.

Dispersing nanoparticles using water produced a detrimental effect on the mechanical and curing properties of the composites. On the other hand, the composite prepared without water demonstrated improvements in the physical properties of the composite induced by the filler. Nanoindentation tests performed on all the specimens confirm the beneficial effect of the nanoparticles in the composite and the deleterious impact of water. Moreover, a saturation point of 2 wt. % of nanofillers was found.

The null hypothesis can be rejected according to the obtained results. The overall structural properties are deeply affected by the presence of water and filler.

Different studies claim that discoloration of composite cements used in endodontic procedures is the major limitation for these kinds of materials [35,61,62]. Even if composite loaded with bismuth oxide filler showed negligible discoloration over time [63], a recent study reported that in particular conditions the discoloration in the long period could be significant [64]. The discoloration process was not investigated in this pilot study, focusing the efforts to understand how the filler interacts with the lattice of the cement and the dispersion medium used. Indeed, further studies are needed to evaluate the impact of the filler and water, also, on the discoloration process.

BiONPs in bio-cement are highly promising due to injectability, bioactivity, and biocompatibility. This pilot study suggests that BiONPs application as nanofiller can be further explored for bone tissue and dental applications.

## Figures and Tables

**Figure 1 materials-14-01626-f001:**
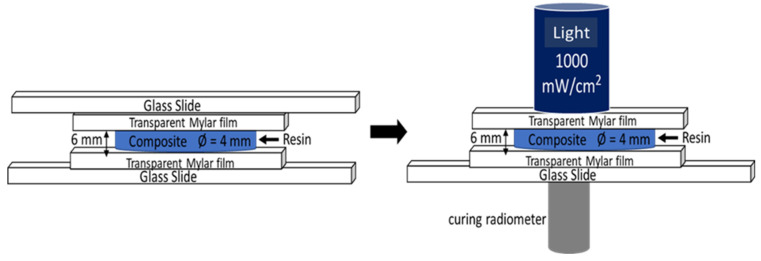
Schematic representation of the depth of cure analysis on composite.

**Figure 2 materials-14-01626-f002:**
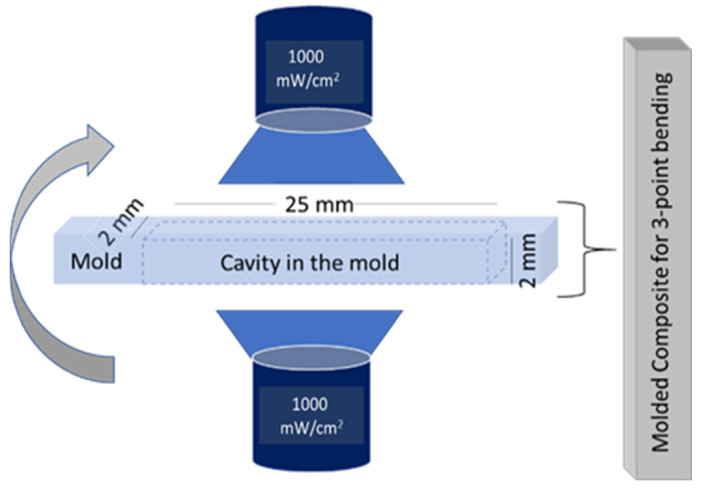
Composite sample curing process.

**Figure 3 materials-14-01626-f003:**
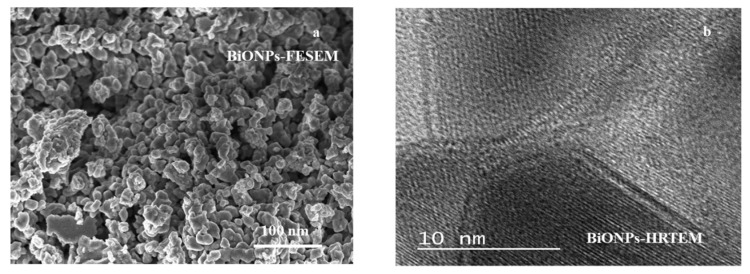
Microscopic analysis of BiONPs (**a**) morphology analysis through the FESEM technique and (**b**) crystal structure through TEM analysis.

**Figure 4 materials-14-01626-f004:**
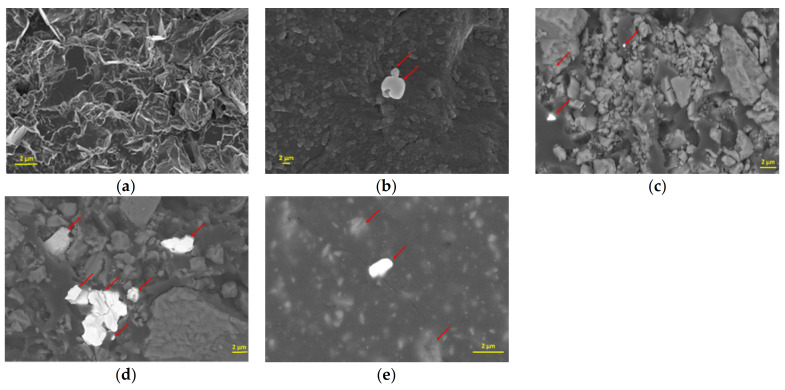
FESEM analysis of (**a**) blank sample. Sample obtained dispersing BiONPs with water in quantity of: (**b**) 1 wt. %; (**c**) 2 wt. %; (**d**) 4 wt. %. (**e**) Sample obtained without water using BiONPs in 2 wt. %. Red arrows indicate the BiONPs in the composite matrix.

**Figure 5 materials-14-01626-f005:**
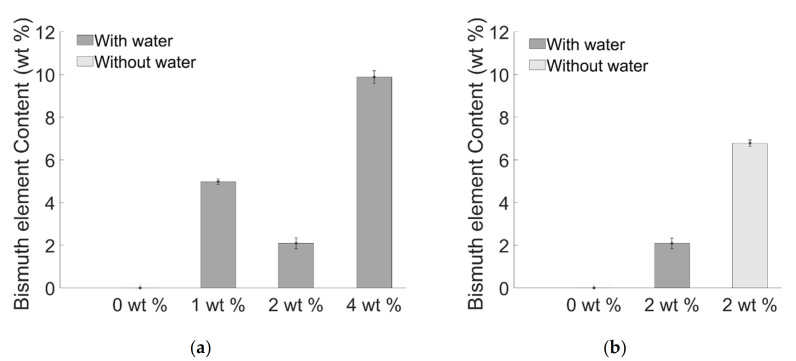
Histograms of EDS analysis on a surface of 4 µm^2^ obtained for sample: (**a**) with water and (**b**) without water.

**Figure 6 materials-14-01626-f006:**
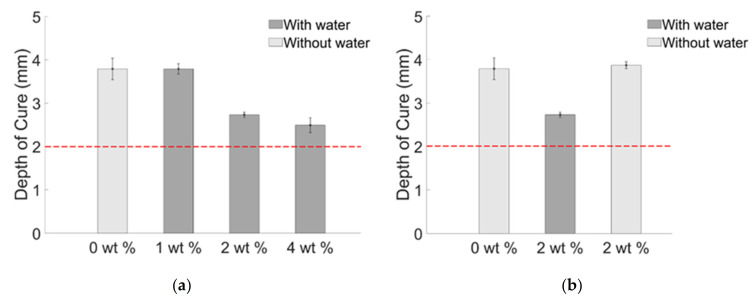
Depth of curing properties of composite (**a**) with water (**b**) with and without water. The red line indicates the minimum curing depth required for in-vivo application.

**Figure 7 materials-14-01626-f007:**
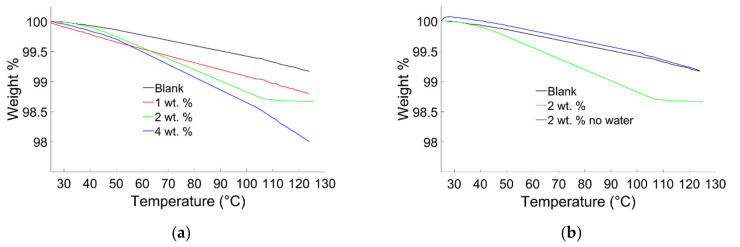
(**a**) Curves of TGA analysis on samples obtained using water. (**b**) Comparison between representative curves of TGA analysis carried on samples obtained with and without water.

**Figure 8 materials-14-01626-f008:**
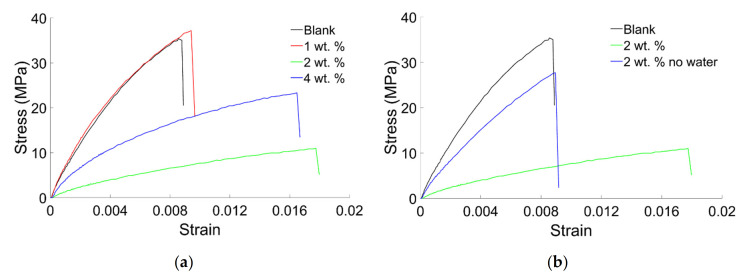
(**a**) Representative curves of three-point bending tests performed on samples obtained using water. (**b**) Comparison between representative curves of three-point bending tests carried on samples obtained with and without water.

**Figure 9 materials-14-01626-f009:**
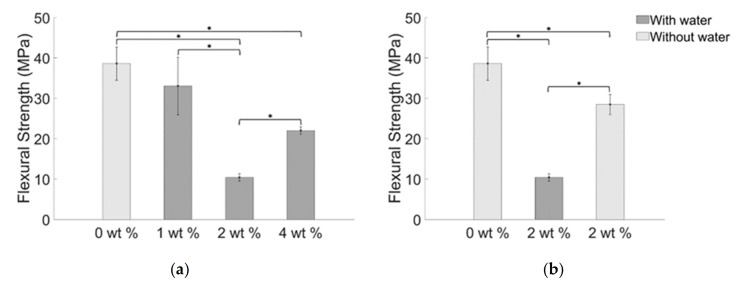
Mean values of flexural strength for BiONP-composite (**a**) with water and (**b**) without water. The bar with the asterisk indicates a statistical difference between the samples (*p* < 0.05).

**Figure 10 materials-14-01626-f010:**
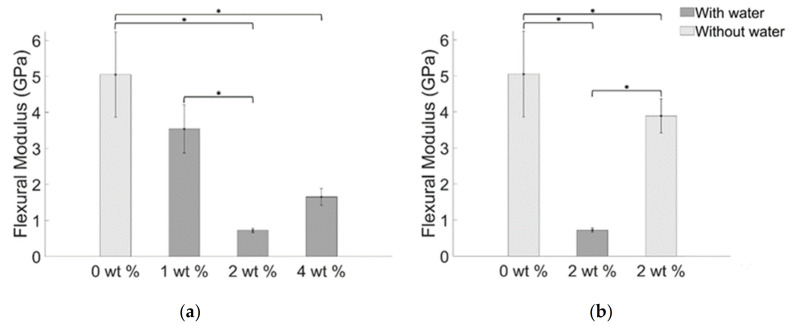
Mean values of flexural modulus for BiONPs—composite (**a**) with water and (**b**) without water. The bar with the asterisk indicates a statistical difference between the samples (*p* < 0.05).

**Figure 11 materials-14-01626-f011:**
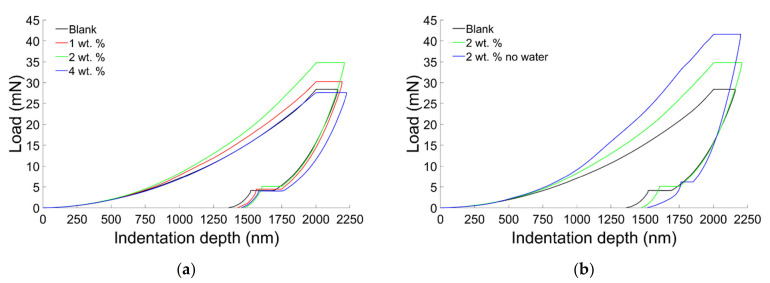
(**a**) Representative curves of nanoindentation tests performed on samples obtained using water. (**b**) Comparison between representative curves of nanoindentation tests carried on samples obtained with and without water.

**Figure 12 materials-14-01626-f012:**
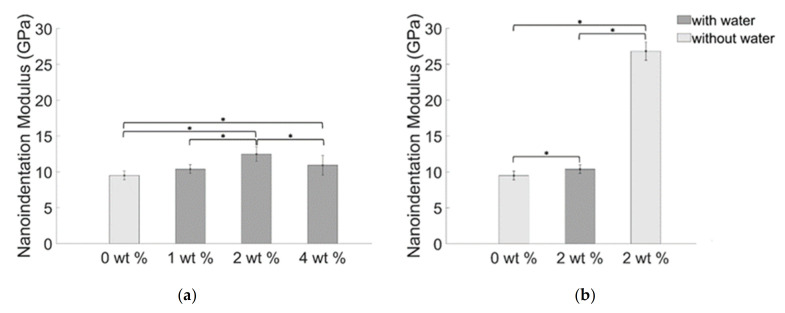
Average values of the nanoindentation modulus of BiONPs composite: (**a**) with water and (**b**) without water. Statistically significant differences (*p* < 0.05) between cements with different filler concentration are also indicated through the asterisk.

**Figure 13 materials-14-01626-f013:**
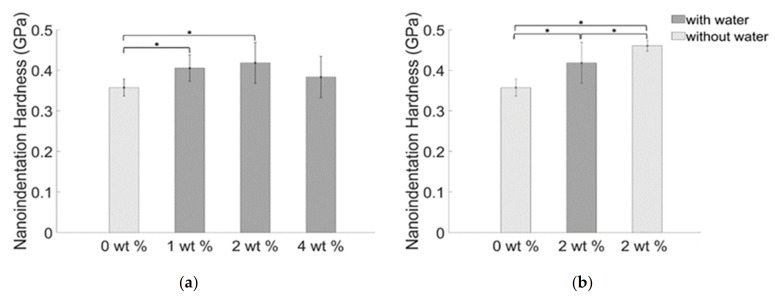
Average values of the nanoindentation hardness for BiONPs—composite (**a**) with water and (**b**) without water. Statistically significant differences (*p* < 0.05) between cements with different filler concentration are also indicated through the asterisk.

**Table 1 materials-14-01626-t001:** Number of specimens and content of inorganic elements. The asterisk indicates that the sample was obtained without using water.

Sample	SiO_2_ (g)	BiONP (g)	Water (g)	Number of Specimens
0 wt. %	35.58	-	-	10
1 wt. %	35.23	0.35	1.05	10
2 wt. %	34.88	0.7	2.1	10
2 wt. % *	34.88	0.7	-	10
4 wt. %	34.18	1.4	4.2	10

**Table 2 materials-14-01626-t002:** Component details used for sample preparation.

Chemical Name	Code	Producer	Quantity (g)
Bisphenol-A glycerolate dimethacrylate (98%)	BisGMA	Sigma-Aldrich (St. Louis, MO, USA)	10.50
Tri(ethylene glycol) di-methacrylate (95%)	TEGDMA	Sigma-Aldrich (St. Louis, MO, USA)	4.50
1,2-(−)-Camphor quinone (99%)	CQ	Sigma-Aldrich (St. Louis, MO, USA)	0.05
N, N-Dimethyl-p-toluidine (99%)	DMPT	Sigma-Aldrich (St. Louis, MO, USA)	0.20
Silica dioxide	SiO_2_	Sigma-Aldrich (St. Louis, MO, USA)	35.58
Bismuth oxide nanoparticles	BiONP	POLITO (Turin, Italy)	0.35/0.7/1.4
Water	DM water	POLITO (Turin, Italy)	1.05/2.1/4.2

## Data Availability

Data is contained within the article.

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
