# Peer review of "Physical Characterization of Bismuth Oxide Nanoparticle Based Ceramic Composite for Future Biomedical Application"

_materials, 2021, doi:10.3390/ma14071626_

Round 1
Reviewer 1 Report
This research is under the scope of this journal; the topic is interesting for readers and this research deals with potentially significant knowledge to the field.
However, there are some concerns about the present manuscript:
Title
- Also, identify the studied in vitro
Abstract
- How many samples? Identified in the abstract.
- In the results, is important to show more information, add some of the p-values.
Introduction
- Page 2 line 31 in dentistry, the oxide of bismuth was used as a contrast product in Calcium silicate cement (CSC materials) with many applications in endodontic filed, like apexification, and pulp regeneration. Please, add more a reference an on this sentence to support these procedures, https://doi.org/10.3390/app9193942.
- Why the authors call eco-friendly? Please Justified for the readers.
- What was the null hypothesis for this study? Add on the last sentence in the introduction and reject in the discussion
Materials and Methods
- Add a table with the groups, Number of samples per group, material and lot Add to table 1.
- What was the manufacturer's instructions for these materials? Add on table materials.
- How many samples for groups? How was the sample calculated? Did the authors perform power analysis to evaluate if this sample size was appropriate?
- When mentioning materials or devices: for some of them you don't mention the manufacturer at all, for some you mention only the manufacturer, for some the manufacturer and city, for some you mention the manufacturer and city/ country.
Results
- Figures need to improve resolution and size.
The font in the graphics is different from the text. Please, standardized the sized and the font in the figures and charts with the font of the manuscript.
- Make other analyze of colour variation (∆E) over time considering only the materials, independent of the irrigant used. And other analyze of colour variation (∆E) over time considering only treatment (Irrigant) variation independent of the material used.
Discussion
- Please, clarified what was the limitation of this study? And also, future perspectives.
One of the big problems of the contrast agent (oxide bismuth) on CSC is teeth discolouration with the time! references used these (10.1111/j.1365-2591.2012.02053.x https://doi.org/10.3390/app10175793; https://doi.org/10.3390/jfb10010014; https://doi.org/10.1371/journal.pone.0240634
Perhaps in the future, the authors must analyze the colour changes on these nanomaterials. Because the discolouration is more marked in a medium and long time of evaluation. https://doi.org/10.1371/journal.pone.0240634 ; https://doi.org/10.1007/s00784-012-0794-1
References
- Check reference’s format MDPI in the manuscript, and in the references. The titles of references have a different format, the title of the article is written in capital letters at the beginning of words, others only in lower case. Also, the standardized format of presentation in the journal's name. Because names have written in a different format, one is not abbreviated, others are not.
Reviewer 2 Report
In this manuscript (materials-1127005), authors have used water as dispersion media for Bismuth oxide nanoparticles and demonstrated the effect on composite properties for medical applications, and characterized by some physical properties (e.g., SEM, mechanical, and thermal).
After going through the manuscript, I do not find much novelty in this work. This study is very simple process to be observed.
As authors claiming this material for medical applications (i.e. bone and dental), they should have designed this study by keeping this parameter for deeper investigation, such as structural analyses (e.g, FTIR, XRD, SEM-EDX), cell studies, etc.
Also, actual plots of mechanical characterization alongwith digital images of BiONPs-based composites should be provided for better understanding of this material in the composite system.
In this current state, I would not recommend this article to be published in this journal.
Round 2
Reviewer 1 Report
This research is under the scope of this journal; the topic is interesting for readers and this research deals with potentially significant knowledge to the field and an open new way for future studies.
The authors improved the quality of the manuscript after the reviewer's indications.
Reviewer 2 Report
Authors have now responded well against the Reviewer comments and improved the manuscript accordingly. In my opinion, this manuscript now can be accepted for publication.